# FEATURE SYNERGY AND INTERFERENCE: AN ANALYSIS FOR TIME-SERIES CLASSIFICATION

## ABSTRACT

The pursuit of a universal, one-size-fits-all model has dominated Time Series Classification (TSC) research. This work challenges that paradigm, arguing that advancing TSC requires a fundamental understanding of feature interplay, not merely more complex architectures. We conduct a series of meticulously designed controlled experiments to dissect the feature spaces of a wide array of representative TSC models, from efficient feature extractors like ROCKET to state-of-the-art deep learning architectures including Transformers and Mamba. For high-dimensional feature extractors, we reveal that the performance bottleneck is dataset-dependent, shifting between feature redundancy and feature noise. We demonstrate that for complex non-linear classifiers, feature pruning can serve as a critical de-noising step on noisy datasets, while for simpler linear models, the full feature set can sometimes be more robust. For a diverse set of nine deep models, we systematically evaluate time-frequency fusion strategies, showing that the optimal choice is intricately linked to both the dataset's intrinsic properties and the model's architectural biases. We uncover clear and widespread evidence of "feature synergy", where fusion provides significant gains, and "feature interference", where it actively degrades performance. Our work pivots the focus from a "model-centric" to a "feature-centric" perspective, providing a new paradigm and a concrete analytical framework for developing adaptive and truly robust TSC solutions.

## 1 INTRODUCTION

Time Series Classification (TSC) is a critical task in diverse domains, from medical diagnosis to industrial monitoring (Fawaz et al., 2019). Research in TSC has largely bifurcated into two streams: highly efficient methods that transform time series into a feature space, exemplified by ROCKET (Dempster et al., 2020), and a vast array of deep learning models that learn representations end-to-end, spanning from foundational CNNs like FCN and ResNet (Wang et al., 2017) to modern architectures like Transformers (Vaswani et al., 2017) and State Space Models (Gu & Dao, 2023).

Despite remarkable progress, the field operates under a persistent "one-size-fits-all" assumption, where new models are benchmarked across extensive archives (Dau et al., 2019) with the implicit goal of achieving universal superiority (Bagnall et al., 2017). However, this approach often yields unstable, dataset-dependent performance rankings. Our preliminary attempts to enhance deep models with spectral features yielded inconsistent results, prompting a more fundamental inquiry: Do we truly understand the features these models generate and how they should be combined?

This paper argues that the prevailing model-centric view obscures a more critical underlying issue: the complex, often counter-intuitive interactions between features, both internal and external to the model. We shift to a feature-centric analysis, a perspective gaining traction where abstracting features like time-series shapes as tokens has shown promise (Wen et al., 2024). Our goal is to investigate the conditions that lead to feature "synergy" versus feature "interference" across a wide range of models and datasets. We not only demonstrate these phenomena but also provide quantitative evidence linking them to intrinsic dataset properties. This leads us to our core research questions:

- **RQ1: For high-dimensional feature extractors like ROCKET and its variants, is the primary performance bottleneck simple feature redundancy, or a more complex interaction between feature noise and the capacity of the downstream classifier?**
- **RQ2: Across a diverse range of deep learning architectures (CNNs, RNNs, Transformers, SSMs), is there a universally optimal strategy for fusing time-domain and frequency-domain features? If not, can we demonstrate that the choice between synergy and interference is predictably linked to the interplay between model architecture and dataset characteristics?**

To answer these questions, we conduct two comprehensive experimental studies (Task 1 and Task 2). Our contributions are threefold:

1. We demonstrate that the performance bottleneck in ROCKET-like feature spaces is a complex interaction between feature properties and classifier capacity, showing that feature pruning can serve as a vital de-noising step for high-capacity models on noisy datasets.

2. We provide the first systematic, comparative analysis of time-frequency fusion strategies across nine distinct deep learning architectures, proving the optimal strategy is dataset- and model-dependent and uncovering compelling evidence of both "feature synergy" and "feature interference".

3. We lay the groundwork for a new, adaptive paradigm in TSC by showing that these feature-centric phenomena are not random, but are linked to intrinsic properties of the data, suggesting a path towards automated strategy selection.

## 2 RELATED WORK

Our research is positioned at the intersection of three key areas: efficient feature-based TSC, deep learning-based TSC, and feature fusion techniques.

### 2.1 EFFICIENT TIME SERIES CLASSIFICATION

The first stream of research focuses on transforming time series into a feature space amenable to fast and robust classifiers. Methods range from dictionary-based approaches like BOSS (Schäfer, 2015) and shapelet-based models (Ye & Keogh, 2011) to complex transformation-based ensembles like HIVE-COTE (Bagnall et al., 2016). A significant breakthrough in this area is ROCKET (Dempster et al., 2020), which generates a large number of features from random convolutional kernels. ROCKET and its variants, such as MINIROCKET (Dempster et al., 2021), achieve state-of-the-art accuracy with remarkable computational efficiency. However, this efficiency comes at the cost of generating a very high-dimensional feature space (typically 10,000 features), which is largely treated as a black box. While their effectiveness is undisputed, the nature of this feature space remains a topic of active research. The challenge of managing its high dimensionality, particularly in terms of feature noise and redundancy, has been acknowledged in recent studies exploring sequential feature selection for these random kernels (Uribarri et al., 2024). This motivates our Task 1 experiments, which aim to systematically dissect these properties.

### 2.2 DEEP LEARNING FOR TIME SERIES CLASSIFICATION

The second stream leverages deep neural networks to learn hierarchical representations directly from raw time series data. Foundational work adapted architectures from computer vision, establishing strong baselines with Fully Convolutional Networks (FCN) and Residual Networks (ResNet) (Wang et al., 2017), and later achieving top-tier performance with models like InceptionTime (Fawaz et al., 2020). Concurrently, Recurrent Neural Networks (RNNs) like LSTM (Hochreiter & Schmidhuber, 1997) and GRU were applied to capture temporal dependencies.

More recently, the field has been influenced by progress in other sequence modeling domains. Transformer-based models (Vaswani et al., 2017), often drawing inspiration from computer vision (Ni et al., 2025), have been successfully adapted for time series. A prominent approach involves treating time series "patches" as tokens, as popularized by PatchTST (Nie et al., 2023). This patch-based paradigm is an area of active research, with recent studies exploring advanced pre-training

strategies, architectural variations, and novel loss functions (Woo et al., 2024; Lu et al., 2025; Yu et al., 2025). As an alternative to attention, State Space Models (SSMs), extensively reviewed in (Somvanshi et al., 2025), have emerged as a powerful new paradigm. The Mamba architecture (Gu & Dao, 2023), in particular, has spurred a wave of research into new time series applications and variants (Ye, 2025; Somvanshi et al., 2024) due to its linear-time complexity and impressive performance on long sequences. Our study is the first to systematically analyze feature fusion across a wide gamut of architectures, from foundational CNNs to modern Transformers and SSMs. Our model zoo also includes other notable CNN architectures like OmniScaleCNN, and the explainable XCM model.

### 2.3 Feature Fusion in Machine Learning

Feature fusion is a long-standing topic in multi-modal learning, where the goal is to combine information from different sources (Baltrušaitis et al., 2018; Ngiam et al., 2011). Common strategies range from simple concatenation and element-wise addition to more complex mechanisms like gating (Hochreiter & Schmidhuber, 1997) and bilinear pooling (Lin et al., 2015; Zhang et al., 2017). Gating mechanisms allow features from one modality to modulate another, while bilinear models capture pairwise interactions between all feature dimensions. While these fusion methods are well-studied in other domains like multimodal learning (Baltrušaitis et al., 2018) and biomedical signal processing (Wang et al., 2024), their application in TSC for combining temporal and spectral features has been ad-hoc. The challenge of learning effective feature representations is, in fact, a central theme in modern time-series analysis (Trirat et al., 2024). Our work fills a critical gap by rigorously evaluating core fusion strategies across nine deep learning architectures, providing a needed systematic analysis of their effectiveness and limitations.

## 3 Methodology: A Framework for Feature Dissection

To systematically investigate our research questions, we designed two comprehensive experimental tasks. Let a given dataset be a collection of time series samples $\mathcal{D} = \{(X_i, y_i)\}_{i=1}^N$, where $X_i \in \mathbb{R}^{L \times C}$ is a multivariate time series of length $L$ with $C$ channels, and $y_i$ is its corresponding class label. All experiments were repeated five times with different random seeds, and results are reported as mean $\pm$ std.

### 3.1 Task 1: Analysis of High-Dimensional Feature Spaces

This experiment investigates the properties of feature sets generated by ROCKET-like methods. Let $\Phi : \mathbb{R}^{L \times C} \to \mathbb{R}^D$ be a feature extraction function that maps a time series sample $X_i$ to a $D$-dimensional feature vector $F_i$. We employ two such functions: $\Phi_{\text{Py-ROCKET}}$ which yields $D = 20,000$ features, and $\Phi_{\text{sk-MINIROCKET}}$ which yields $D = 10,000$ features.

For a given feature set $\{F_i\}$, we evaluate its quality using a classifier $C$, chosen from a non-linear LightGBM ($C_{\text{LGBM}}$) and a linear RidgeClassifierCV ($C_{\text{Ridge}}$). We test three feature processing strategies:

1. **Base**: The classifier is trained directly on the full feature set, $C(F_i)$.

2. **Pruned**: A supervised selection strategy. A subset of $k = 500$ features $F_i' \subset F_i$ is selected by choosing the features with the highest ANOVA F-statistic scores on the training data. The classifier is then trained on this subset, $C(F_i')$.

3. **PCA**: An unsupervised reduction strategy. A projection matrix $W \in \mathbb{R}^{D \times k}$ with $k = 500$ is learned from the training data via Principal Component Analysis. The classifier is trained on the projected features, $C(W^T F_i)$.

To ground these results, we compare against a **Barycenter-DTW** baseline. For each class $c$, a barycenter (average time series) $\bar{X}_c$ is computed from the training data using DTW barycenter averaging. A test sample $X_{test}$ is then classified as $\arg\min_c \text{DTW}(X_{test}, \bar{X}_c)$.

## 3.2 Task 2: Analysis of Deep Learning Fusion Strategies

This experiment investigates feature synergy and interference across nine deep learning architectures, denoted generically as $M$. The model set includes CNNs (InceptionTime, ResNet, FCN, OmniScaleCNN, XCM), RNNs (LSTM, GRU), and modern architectures (PatchTST, Mamba).

The core of this task is to compare the model's baseline performance, a mapping $M : \mathbb{R}^{L \times C} \to \mathbb{R}^{N_{classes}}$, against its performance when features are fused. Let $M_\theta$ be a model with parameters $\theta$. We define the time-domain features $F_{time} \in \mathbb{R}^{d_t}$ as the output of its penultimate layer, $M_{\theta, l-1}(X_i)$, after being trained on the time series data alone. The frequency-domain features $F_{spec} \in \mathbb{R}^{d_s}$ are derived from a Wavelet Transform, where the $d_s = 50$ most informative frequency bands are selected via ANOVA F-test.

We evaluate two canonical fusion strategies, where a small classification head $h : \mathbb{R}^{d_{final}} \to \mathbb{R}^{N_{classes}}$ is trained on the fused features. Let $F_{time}$ and $F_{spec}$ be the feature vectors for a sample.

- **Concat Fusion**: The final feature vector is the concatenation, $F_{final} = [F_{time}; F_{spec}] \in \mathbb{R}^{d_t + d_s}$. The model computes $h(F_{final})$.

- **Gating Fusion**: The spectral features modulate the temporal features. The final representation is $F_{final} = F_{time} \odot \sigma(\text{MLP}(F_{spec}))$, where $\odot$ denotes element-wise multiplication, $\sigma$ is the sigmoid function, and an MLP aligns the feature dimensions. The model computes $h(F_{final})$.

The performance of these strategies is compared against the **Time-Only (Base)** performance, which is the accuracy achieved by the fully-trained end-to-end model $M_\theta(X_i)$.

## 4 Experiment Results and Analysis

Our extensive experiments, designed to be both broad and deep, yield several key insights into the nature of time series features and their interactions across a wide range of models. We present the main results for Task 1 (high-dimensional feature spaces) and Task 2 (deep learning fusion) in Table 1 and Table 2, respectively. All results are reported as the mean accuracy (%) $\pm$ standard deviation over five independent runs with different random seeds.

### 4.1 Task 1: The Classifier-Feature Interplay in High Dimensions

Our first set of experiments dissects the feature spaces of ROCKET-like extractors. Figure 1 provides a detailed visual case study on the ExchangeRate dataset, which succinctly summarizes the key finding of this section: the performance of a feature strategy is critically dependent on the capacity of the downstream classifier. The full results, presented in Table 1, further substantiate this conclusion across multiple datasets and feature extractors.

The central finding from this task is the significant interaction between the feature set and the classifier's capacity, a phenomenon clearly illustrated for the ExchangeRate dataset in Figure 1. For the high-capacity, non-linear **LightGBM** classifier, both supervised pruning (*Pruned (sup.)*) and our proposed unsupervised method (*Pruned-KMeans (unsup.)*) dramatically outperform the baseline. This is particularly evident with Py-ROCKET features, where the unsupervised method (70.54%) approaches supervised performance (74.05%) and is a marked improvement over the baseline (65.95%). This demonstrates that on noisy data, the performance bottleneck is indeed significant "feature noise", which can be effectively mitigated to unlock performance gains, even without access to labels.

Conversely, the simpler linear **RidgeClassifierCV** often shows different preferences. On the ETTh1 dataset, this lower-capacity model achieves its best performance with the aggressive, unsupervised dimensionality reduction of **PCA** (45.81%), suggesting that creating a dense, decorrelated representation is more beneficial than navigating the original, noisy feature space for a simpler model.

Finally, the **Barycenter-DTW** baseline provides a crucial reference point. Its outstanding performance on ExchangeRate (77.03%) suggests this dataset's classes are well-separated by overall time

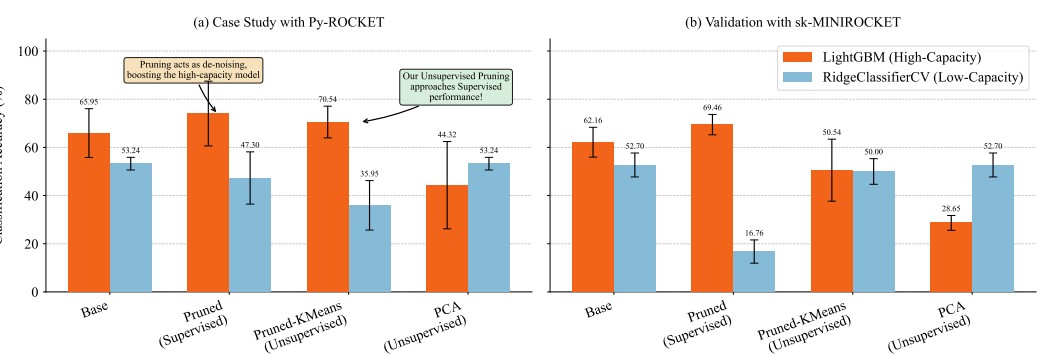

Figure 1: A visual case study on the 'ExchangeRate' dataset illustrating the classifier-feature interplay. **(a)** Using 'Py-ROCKET' features, the high-capacity LightGBM model clearly benefits from both supervised and our proposed unsupervised pruning, which act as de-noising mechanisms. **(b)** The same general trend is validated using 'sk-MINIROCKET' features, demonstrating the robustness of this finding across different feature extractors.

Table 1: Results for Task 1: Analysis of High-Dimensional Feature Spaces. We report mean classification accuracy (%) ± std over 5 runs. The table includes both supervised (*Pruned (sup.)*) and our proposed unsupervised (*Pruned-KMeans (unsup.)*) pruning strategies. Best performance for each dataset is in **bold**.

| Classifier | Feature Extractor | Strategy | ETTh1 | ETTm1 | ExchangeRate | Weather |
|---|---|---|---|---|---|---|
| LightGBM | Py-ROCKET | Base | 30.81 ± 4.03 | 32.69 ± 1.28 | 65.95 ± 10.13 | 47.58 ± 2.67 |
| | | Pruned (sup.) | 29.77 ± 7.71 | 23.77 ± 1.84 | 74.05 ± 13.43 | 44.53 ± 0.98 |
| | | Pruned-KMeans (unsup.) | 30.58 ± 4.36 | 33.01 ± 0.68 | 70.54 ± 6.58 | 42.48 ± 2.03 |
| | | PCA (unsup.) | 31.86 ± 3.80 | 33.38 ± 1.07 | 44.32 ± 18.12 | 37.30 ± 2.09 |
| | sk-MINIROCKET | Base | 39.88 ± 7.91 | 28.78 ± 1.77 | 62.16 ± 6.19 | **55.62 ± 2.32** |
| | | Pruned (sup.) | 41.86 ± 2.70 | 36.92 ± 1.42 | 69.46 ± 4.23 | 34.13 ± 1.73 |
| | | Pruned-KMeans (unsup.) | 36.16 ± 3.52 | 32.89 ± 1.31 | 50.54 ± 12.88 | 52.38 ± 2.30 |
| | | PCA (unsup.) | 27.44 ± 3.62 | 31.42 ± 1.01 | 28.65 ± 3.08 | 49.90 ± 2.19 |
| RidgeClassifierCV | Py-ROCKET | Base | 38.37 ± 1.48 | 35.25 ± 4.06 | 53.24 ± 2.63 | 45.37 ± 2.50 |
| | | Pruned (sup.) | 36.98 ± 4.30 | 32.46 ± 5.12 | 47.30 ± 10.85 | 40.34 ± 3.72 |
| | | Pruned-KMeans (unsup.) | 35.00 ± 1.95 | **38.79 ± 6.87** | 35.95 ± 10.27 | 42.13 ± 2.78 |
| | | PCA (unsup.) | 37.67 ± 0.96 | 30.71 ± 1.18 | 53.24 ± 2.63 | 46.10 ± 1.58 |
| | sk-MINIROCKET | Base | 45.70 ± 1.40 | 31.42 ± 0.72 | 52.70 ± 4.97 | 50.32 ± 0.53 |
| | | Pruned (sup.) | 44.07 ± 2.61 | 29.41 ± 1.89 | 16.76 ± 4.83 | 35.81 ± 2.02 |
| | | Pruned-KMeans (unsup.) | 43.72 ± 2.30 | 31.60 ± 3.71 | 50.00 ± 5.32 | 49.45 ± 1.57 |
| | | PCA (unsup.) | **45.81 ± 0.76** | 34.88 ± 1.05 | 52.70 ± 4.97 | 49.75 ± 1.24 |
| Barycenter-DTW | Raw-DTW | N/A | 31.40 ± 0.00 | 26.33 ± 0.00 | **77.03 ± 0.00** | 26.86 ± 0.00 |

series shape. Its weaker performance on datasets like Weather (26.86%) confirms that other datasets require the local, nuanced patterns that only convolutional features like ROCKET can capture. This directly addresses **RQ1**: the bottleneck in ROCKET's feature space is a complex interplay between feature properties and classifier capacity, with the optimal resolution depending on intrinsic dataset characteristics.

## 4.2 TASK 2: THE FALLACY OF UNIVERSAL FUSION - SYNERGY VS. INTERFERENCE

The results from our large-scale deep learning experiments, presented in Table 2, provide a clear and compelling answer to our second research question (RQ2): there is no universally optimal fusion strategy. The decision to fuse, and the choice of fusion method, is deeply contingent on both the model architecture and the dataset.

The phenomena of "feature synergy" and "feature interference" are widespread. On the **Weather** dataset, for instance, we observe consistent **synergy**; nearly all architectures, including ResNet, FCN, and OmniScaleCNN, achieve higher accuracy with fusion than in their Time-Only setting. This suggests the spectral features provide complementary information that the time-domain models alone cannot capture. Conversely, on the **ExchangeRate** dataset, we witness stark **interference**.

Table 2: Results for Task 2: Deep Learning Fusion Strategy Analysis. We report mean classification accuracy (%) ± std over 5 runs. Best performance for each dataset and model architecture is in **bold**.

| Model Architecture | Feature Input | ETTh1 | ETTm1 | ExchangeRate | Weather |
|---|---|---|---|---|---|
| FCN | Concat Fusion | 35.23 ± 7.66 | 43.08 ± 3.57 | 14.59 ± 10.66 | 53.22 ± 2.69 |
| | Gating Fusion | 37.44 ± 4.91 | 41.29 ± 1.41 | 22.43 ± 5.20 | 52.99 ± 2.81 |
| | Time-Only (Base) | **38.02 ± 11.78** | **41.70 ± 4.00** | **21.08 ± 2.26** | **52.69 ± 2.87** |
| GRU | Concat Fusion | 28.26 ± 3.23 | 40.95 ± 5.59 | 51.89 ± 25.97 | 60.04 ± 1.29 |
| | Gating Fusion | 29.77 ± 3.20 | 41.41 ± 3.92 | **67.30 ± 6.98** | 60.57 ± 1.10 |
| | Time-Only (Base) | **30.12 ± 2.86** | **41.53 ± 3.67** | 67.03 ± 4.34 | **60.84 ± 0.84** |
| InceptionTime | Concat Fusion | 36.51 ± 5.72 | 34.96 ± 9.93 | 42.97 ± 19.87 | 56.61 ± 4.23 |
| | Gating Fusion | 34.77 ± 7.88 | 35.57 ± 5.32 | **63.78 ± 13.56** | **58.59 ± 2.17** |
| | Time-Only (Base) | **37.09 ± 14.16** | **36.49 ± 8.41** | 48.65 ± 22.43 | 57.90 ± 4.16 |
| LSTM | Concat Fusion | 28.95 ± 5.22 | 39.48 ± 1.95 | 42.16 ± 17.91 | 56.65 ± 1.70 |
| | Gating Fusion | 28.49 ± 4.92 | 36.95 ± 2.51 | **69.19 ± 8.41** | 56.95 ± 2.68 |
| | Time-Only (Base) | **27.91 ± 5.96** | **37.50 ± 1.74** | 67.84 ± 7.55 | **57.26 ± 3.03** |
| Mamba | Concat Fusion | **22.56 ± 20.77** | **24.32 ± 13.86** | 0.00 ± 0.00 | 36.19 ± 0.00 |
| | Gating Fusion | 13.95 ± 12.92 | 22.30 ± 12.98 | 0.00 ± 0.00 | 36.19 ± 0.00 |
| | Time-Only (Base) | 8.02 ± 11.81 | 17.84 ± 13.09 | 0.00 ± 0.00 | 36.19 ± 0.00 |
| OmniScaleCNN | Concat Fusion | 40.47 ± 11.89 | 44.26 ± 6.83 | 36.49 ± 11.66 | 46.02 ± 5.13 |
| | Gating Fusion | 39.88 ± 2.49 | **47.14 ± 2.61** | 27.30 ± 7.73 | **48.42 ± 3.45** |
| | Time-Only (Base) | **41.05 ± 6.39** | 43.34 ± 5.51 | **24.32 ± 6.55** | 44.00 ± 11.91 |
| PatchTST | Concat Fusion | 35.12 ± 2.98 | 37.09 ± 2.21 | **41.62 ± 9.28** | 46.06 ± 3.22 |
| | Gating Fusion | **40.70 ± 3.41** | 36.32 ± 2.46 | 24.32 ± 4.58 | 44.99 ± 2.99 |
| | Time-Only (Base) | 37.33 ± 2.38 | **36.06 ± 2.72** | 11.35 ± 1.21 | **44.30 ± 3.37** |
| ResNet | Concat Fusion | 30.58 ± 4.80 | 33.27 ± 6.98 | **35.68 ± 15.13** | 58.63 ± 1.87 |
| | Gating Fusion | **32.56 ± 7.68** | **34.99 ± 4.04** | 23.51 ± 10.05 | 58.63 ± 1.17 |
| | Time-Only (Base) | 30.23 ± 9.95 | 34.47 ± 5.85 | 28.65 ± 25.09 | **58.48 ± 2.41** |
| XCM | Concat Fusion | 39.77 ± 7.63 | 32.06 ± 4.04 | 46.49 ± 16.54 | 49.79 ± 1.90 |
| | Gating Fusion | 39.42 ± 2.51 | **37.47 ± 2.88** | **55.68 ± 19.94** | 45.33 ± 2.10 |
| | Time-Only (Base) | **43.37 ± 3.62** | 34.45 ± 3.15 | 37.03 ± 4.54 | **49.98 ± 2.87** |

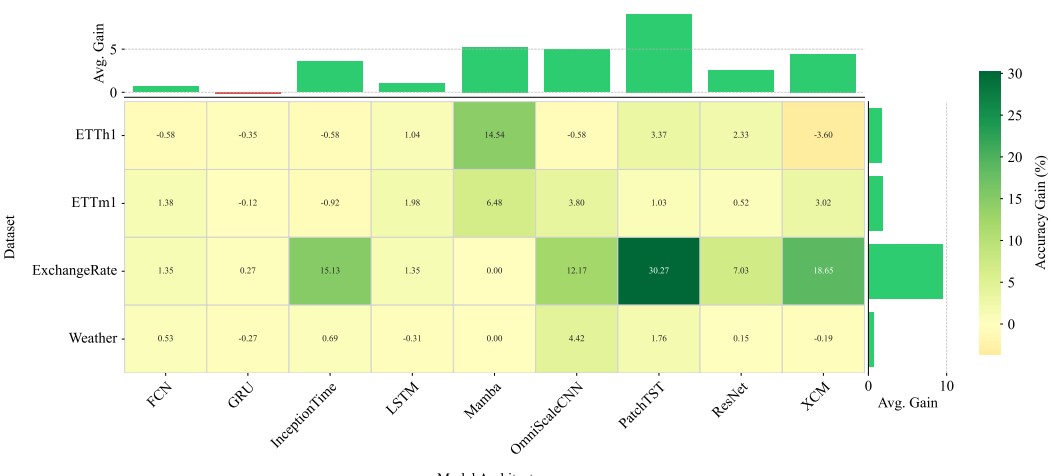

Figure 2: Heatmap of performance gain from feature fusion across all deep learning models and datasets. Each cell represents the accuracy change (%) of the best fusion strategy compared to the Time-Only baseline. Green indicates synergy (fusion helps), while red indicates interference (fusion hurts). The marginal bar plots on the top and right summarize the average gain for each model and dataset, respectively, providing a global view of fusion effectiveness.

The powerful recurrent models, GRU and LSTM, perform exceptionally well in their Time-Only configuration (67.03% and 67.84%, respectively), but their performance collapses dramatically with simple Concat Fusion. This indicates that for these models on this dataset, the spectral features act as a source of noise, corrupting the potent representations learned from the temporal domain.

The choice between fusion methods also matters. On ExchangeRate, the XCM model's accuracy soars from 37.03% to 55.68% with Gating Fusion, demonstrating the gate's ability to selectively filter and apply spectral information. For LSTM on the same dataset, Gating Fusion (69.19%) is also clearly superior to the destructive Concat Fusion (42.16%). This confirms that different model architectures have vastly different abilities to handle and integrate external feature sources, and the fusion mechanism itself is a critical factor in determining the outcome. A notable outlier is our Mamba implementation, which consistently failed to learn on most datasets, suggesting that its base architecture may require significant task-specific hyperparameter tuning beyond the scope of this study.

## 5 Correlating Performance with Dataset Properties

Having established *that* the performance of feature handling strategies is highly contingent on the model and dataset, we now seek to understand *why*. To bridge the gap from qualitative observation to quantitative evidence, we investigate the relationship between intrinsic dataset properties and the success of our feature-handling strategies. This analysis provides strong quantitative backing for our core feature-centric hypotheses.

### 5.1 Meta-Feature Calculation

We compute two meta-features for each dataset based on its raw time series data.

**Signal-to-Noise Ratio (SNR)** is estimated by decomposing each time series into trend, seasonal, and residual components using STL decomposition. SNR is then calculated as the variance ratio of the signal (trend + seasonal) to the noise (residual); a higher SNR indicates a cleaner signal.

**Spectral Entropy** is calculated from the Power Spectral Density (PSD) of the signal, obtained using Welch's method. This metric measures the uniformity of the power distribution across frequencies; a high entropy indicates a complex spectrum.

### 5.2 Qualitative Visual Inspection

To provide an intuitive, visual grounding for these meta-features, Figure 3 contrasts real data samples from our key case study datasets. The top panels clearly illustrate the concept of SNR: Figure 3(a) shows a sample from ExchangeRate, which has a low calculated SNR and appears visually noisy with high-frequency oscillations. In contrast, Figure 3(b) shows a sample from Weather, which has a higher SNR and appears significantly cleaner with more discernible patterns. Similarly, the bottom panels visualize spectral complexity. The power spectrum of ETTm2 (Figure 3(c)), a dataset with low spectral entropy, is dominated by a few distinct frequency peaks. Conversely, the spectrum of Weather (Figure 3(d)), a dataset with high spectral entropy, is more broadly distributed and complex. These visual examples corroborate our quantitative calculations and provide the intuition needed to interpret the following correlation analysis.

### 5.3 Correlation Analysis

Figure 4 plots the performance gains of our strategies against the calculated meta-features, breaking down the analysis for each model combination to verify the generality of our findings. The left panel reveals a consistent negative correlation between a dataset's SNR and the accuracy gain from our 'Pruned' (supervised feature selection) method. For datasets with low SNR (i.e., high noise) like ILI, the gain from pruning is often positive, as it effectively filters out noise. Conversely, for datasets with very clean signals like ExchangeRate, the gain is less pronounced or even negative for some models. This trend holds across different combinations, confirming that the utility of feature pruning is strongly linked to the signal quality.

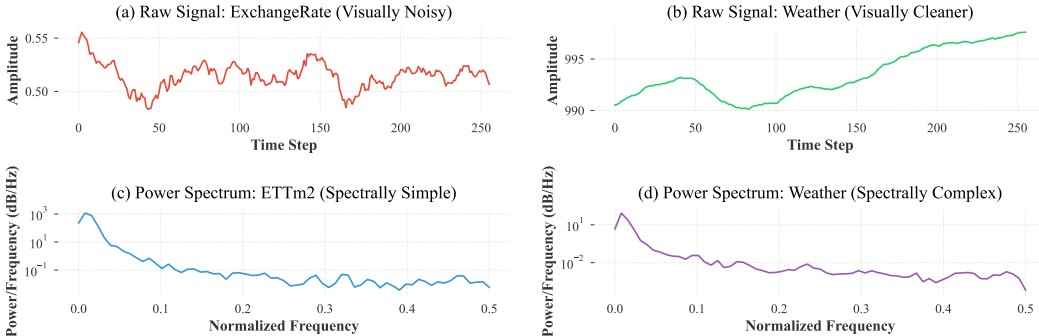

Figure 3: Qualitative inspection of key dataset characteristics using real data samples. **(a) vs (b)** illustrates the visual difference between a noisy signal (ExchangeRate, low SNR) and a cleaner signal (Weather, high SNR). **(c) vs (d)** contrasts a spectrally simple signal (ETTm2, low entropy) with a spectrally complex one (Weather, high entropy). This figure provides visual intuition for the meta-features analyzed in Figure 4.

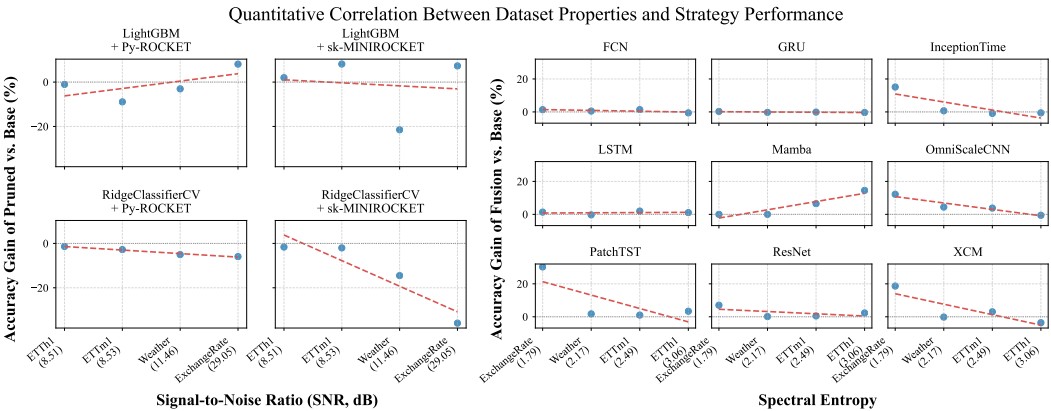

Figure 4: Quantitative correlation between dataset meta-features and strategy performance. **Left (a-d):** The performance gain of the 'Pruned' method is negatively correlated with the dataset's Signal-to-Noise Ratio (SNR). **Right (e-m):** The performance gain of the best fusion strategy is positively correlated with the dataset's Spectral Entropy. This provides strong, multi-model evidence for our feature-centric hypotheses.

The right panel of Figure 4 shows a prevailing positive correlation between a dataset's spectral entropy and the benefit of feature fusion. This trend is visible across a diverse range of architectures. Datasets with high spectral entropy, such as ETTh1, tend to exhibit positive gains from fusion, indicating that their rich frequency-domain information is complementary to the time-domain features. In contrast, datasets with lower spectral entropy, like ExchangeRate and Weather, show more mixed results, where fusion is beneficial for some architectures but detrimental to others. This strongly supports our second hypothesis: the outcome of fusion is not random, but is predictably linked to the spectral complexity of the data and its interaction with the model's architectural biases.

## 6 TOWARDS AN ADAPTIVE FRAMEWORK: A PILOT STUDY

Our analysis consistently suggests that the optimal feature strategy is predictable from a dataset's intrinsic properties. To formally test this hypothesis and demonstrate the feasibility of our proposed adaptive framework, we conducted a pilot study. We created a "hyper-focused" meta-dataset using the results from our most illustrative case: the `LightGBM + Py-ROCKET` combination across our four key datasets. The task for a simple meta-classifier was to predict the binary target—whether

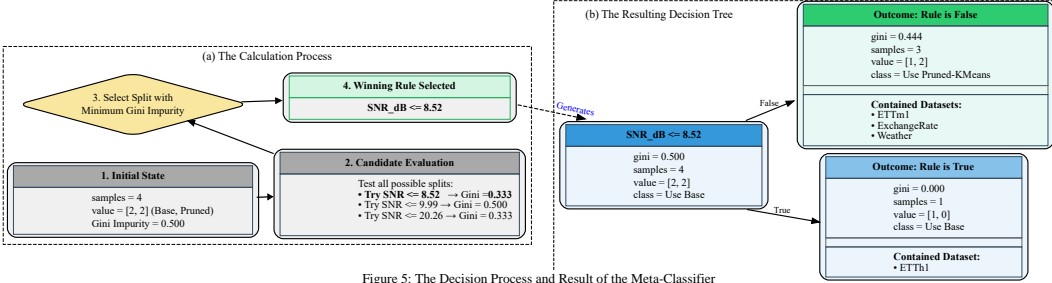

Figure 5: The Decision Process and Result of the Meta-Classifier

Figure 5: Visualizing the meta-classifier's decision process and the final resulting rule. **(a)** A flowchart illustrating how the pilot study evaluates candidate splits based on Gini impurity to select the optimal rule. The algorithm identifies 'SNR ¡= 8.52' as the split that best separates the classes. **(b)** The final, data-rich decision tree generated from this rule. The tree successfully isolates the 'ETTh1' dataset and correctly recommends 'Use Pruned-KMeans' for the branch containing datasets with higher SNR where pruning was shown to be beneficial.

to use our `Pruned-KMeans (unsup.)` strategy (class 1) or the `Base` strategy (class 0)—using only the dataset's SNR and Spectral Entropy as features.

The entire decision process and the final resulting rule from a trained Decision Tree (with `max_depth=1`) are visualized in Figure 5. The process begins with a balanced set of 4 experimental outcomes (2 for each class). As shown in Figure 5(a), the algorithm evaluates all possible splits and determines that a rule based on SNR provides the greatest reduction in impurity. The resulting tree, shown in Figure 5(b), is remarkably effective. It learns a single, intuitive rule, `SNR_dB <= 8.52`, which successfully isolates the `ETTh1` dataset as a case where the base strategy is preferred. More importantly, the other leaf correctly classifies two of the three remaining datasets (`ETTm1`, `ExchangeRate`) as benefiting from our unsupervised pruning method, making `Use Pruned-KMeans` the majority class for that branch.

This successful pilot study provides strong proof-of-concept for an effective, data-driven adaptive approach. It transforms our conceptual framework into a tangible result and lays a clear path for future work in developing fully-automated, adaptive TSC systems.

## 7 DISCUSSION AND CONCLUSION

In this work, we challenged the prevailing model-centric paradigm in Time Series Classification through extensive experiments on both feature-based and a diverse set of nine deep learning models. Our results yield three core contributions: we demonstrated that the bottleneck in high-dimensional feature spaces is a complex interplay between feature noise and classifier capacity; we provided the first large-scale proof that feature fusion can lead to both synergy and interference, a phenomenon predictable from dataset properties like spectral entropy; and through a successful pilot study (Figure 5), we demonstrated the feasibility of an adaptive, feature-centric framework.

The implications of these findings are significant. The interplay between feature pruning and classifier capacity (Figure 1) suggests that feature processing and model selection should be considered a co-design problem. Furthermore, the duality of fusion outcomes (Figure 2) serves as a crucial cautionary tale: feature fusion is a conditional tool, not a universal enhancement, and its success is tied to the alignment of model architecture with data characteristics. This is powerfully illustrated by our Mamba case study: the state-of-the-art model struggled, likely due to a mismatch between its architectural bias (favoring long-range dependencies) and the properties of many TSC datasets. This underscores that even the most advanced models are not "silver bullets" and are subject to the feature-centric principles we advocate for.

While our work provides strong evidence, we acknowledge its limitations: our analysis is focused on classification, and the pilot study, while a successful proof-of-concept, used a small meta-dataset. Ultimately, our work advocates for a paradigm shift: from pursuing a universal model to developing adaptive frameworks that diagnose dataset characteristics and automatically configure optimal analysis pipelines.

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

## ETHICS STATEMENT

This research focuses on a fundamental analysis of feature interactions within established Time Series Classification (TSC) models. All experiments were conducted on publicly available, anonymized benchmark datasets, such as the ETT datasets and the UCR/UEA archives. Our work does not involve human subjects, personally identifiable information, or any form of sensitive data, and therefore raises no direct privacy or security concerns. The insights presented are intended to improve the scientific understanding of TSC models and do not, to the best of our knowledge, have a direct potential for negative societal impact or malicious use. The authors declare no competing interests or conflicts of interest.

## REPRODUCIBILITY STATEMENT

We are committed to ensuring the full reproducibility of our research. All methodologies, parameters, and experimental workflows are detailed extensively between the main paper and this appendix to facilitate the verification of our findings.

- **Datasets and Preprocessing:** All datasets used are public benchmarks. A comprehensive statistical overview is provided in Appendix A.1 (Table 3). The uniform data preprocessing parameters for the sliding window protocol are detailed in Table 4.
- **Experimental Settings:** The specific hyperparameters for all methods in Task 1 (Feature Engineering) and Task 2 (Deep Learning) are exhaustively documented in Appendix A.1, specifically in Table 5 and Table 6, respectively.
- **Algorithms and Methodology:** The high-level workflows for our three core experimental procedures (Task 1, Task 2, and the Pilot Study) are presented as formal pseudocode in Appendix A.2. Algorithm 1, Algorithm 2, and Algorithm 3 correspond to each procedure, detailing the logical steps from data processing to evaluation. These algorithms, in conjunction with the parameters in the preceding tables, provide a complete blueprint for replication.
- **Results:** As stated in Section 4, all reported experimental results are the mean and standard deviation over five runs with different random seeds to ensure the robustness of our conclusions.

## A   APPENDIX

### A.1   DETAILED EXPERIMENTAL SETTINGS

This section provides the detailed configurations and parameters used for all experiments to ensure full reproducibility.

Table 3: Statistics of the datasets used in the experiments.

| Dataset | Train Samples | Test Samples | Dimensions | Classes |
|---------|---------------|--------------|------------|---------|
| ETTh1 | 687 | 172 | 6 | 7 |
| ETTh2 | 687 | 172 | 6 | 7 |
| ETTm1 | 2777 | 695 | 6 | 7 |
| ETTm2 | 2777 | 695 | 6 | 7 |
| Electricity | 5600 | 1401 | 369 | 370 |
| ExchangeRate | 293 | 74 | 7 | 8 |
| ILI | 42 | 11 | 10 | 11 |
| Weather | 2098 | 525 | 20 | 21 |

**Dataset Overview**   Table 3 provides a statistical overview of the public benchmark datasets utilized in our experiments. This includes the number of training and testing samples after applying our sliding window protocol, the dimensionality (number of channels), and the number of unique classes for each dataset.

Table 4: Data preprocessing settings.

| Parameter | Value |
|-----------|-------|
| Sliding Window Size | 256 |
| Stride | 20 |

**Data Preprocessing**   To ensure consistency across all experiments, a uniform data preprocessing pipeline was applied. The key parameters for the sliding window segmentation are detailed in Table 4.

Table 5: Parameters for Task 1 (Feature Engineering Analysis).

| Parameter | Value |
|---|---|
| `Py-ROCKET`: num_kernels | 10,000 (produces 20,000 features) |
| `sk-MINIROCKET`: num_kernels | 10,000 |
| `Pruned (sup.)`: n_features | 500 |
| `Pruned-KMeans (unsup.)`: n_features | 500 |
| `PCA`: n_components | 500 |

**Task 1 Parameters**  The experiments in Task 1 (Section 3) involved several feature extractors and processing strategies. The specific hyperparameters for these methods are listed in Table 5.

Table 6: Hyperparameters for Task 2 (Deep Learning Analysis).

| Parameter | Value |
|---|---|
| Spectral Features: k_bands | 50 |
| Training Epochs (Base Models) | 50 |
| Training Epochs (Fusion Heads) | 30 |
| Batch Size | 32 |
| Learning Rate | 0.001 |
| Optimizer | Adam |

**Task 2 Hyperparameters**  For the deep learning fusion experiments in Task 2, a consistent set of training hyperparameters was used for all nine architectures to ensure a fair comparison of the fusion strategies' effects. These global settings are provided in Table 6.

A.2 CORE ALGORITHM PSEUDOCODE

The following algorithms provide a high-level overview of the experimental workflows, corresponding to the experiments detailed in the main paper.

---

**Algorithm 1** Task 1: Feature Engineering Experiment Workflow

---

1: **Input:** Datasets $\mathcal{D}$, Seeds $S$
2: Initialize empty results list $R_1$
3: **for** each seed in $S$ **do**
4:     **for** each dataset in $\mathcal{D}$ **do**
5:         $(X_{train}, y_{train}), (X_{test}, y_{test}) \leftarrow$ LoadAndProcessData(dataset)
6:         **for** each feature_extractor $\Phi$ in $\{\Phi_{\text{Py-ROCKET}}, \Phi_{\text{sk-MINIROCKET}}\}$ **do**
7:             $F_{train} \leftarrow \Phi(X_{train}); F_{test} \leftarrow \Phi(X_{test})$
8:             **for** each classifier $C$ in $\{C_{\text{LGBM}}, C_{\text{Ridge}}\}$ **do**
9:                 **for** each strategy $\Psi$ in {Base, Pruned, PCA, KMeans-Pruned} **do**
10:                     $F'_{train}, F'_{test} \leftarrow \Psi(F_{train}, F_{test}, y_{train})$
11:                     $M \leftarrow C.\text{fit}(F'_{train}, y_{train})$
12:                     $Acc \leftarrow M.\text{score}(F'_{test}, y_{test})$
13:                     Append $\{seed, dataset, \Phi, C, \Psi, Acc\}$ to $R_1$
14:                 **end for**
15:             **end for**
16:         **end for**
17:     **end for**
18: **end for**
19: **Output:** Results $R_1$

---

**Task 1 Workflow**  Algorithm 1 outlines the complete workflow for our feature engineering experiments (Task 1). The process involves iterating through all seeds, datasets, feature extractors,

classifiers, and strategies to systematically collect performance data and ensure the robustness of our findings.

---

**Algorithm 2** Task 2: Deep Learning Fusion Experiment Workflow

---

1: **Input:** Datasets $\mathcal{D}$, Seeds $S$, Architectures $\mathcal{M}$
2: Initialize empty results list $R_2$
3: **for** each seed in $S$ **do**
4:     **for** each dataset in $\mathcal{D}$ **do**
5:         $(X_{train}, y_{train}), (X_{test}, y_{test}) \leftarrow$ LoadAndProcessData(dataset)
6:         $F_{spec,train}, F_{spec,test} \leftarrow$ ExtractSpectralFeatures$(X_{train}, y_{train}, X_{test})$
7:         **for** each ModelArchitecture in $\mathcal{M}$ **do**
8:             $M_{base} \leftarrow$ ModelArchitecture()
9:             $M_{base}$.train$(X_{train}, y_{train})$                                    ▷ Train end-to-end
10:            $Acc_{base} \leftarrow M_{base}$.evaluate$(X_{test}, y_{test})$
11:            Append $\{..., \text{'Time-Only'}, Acc_{base}\}$ to $R_2$
12:            $F_{time,train} \leftarrow M_{base}$.extract_features$(X_{train})$
13:            $F_{time,test} \leftarrow M_{base}$.extract_features$(X_{test})$
14:            **for** each fusion_strategy in {Concat, Gating} **do**
15:                $h_{fusion} \leftarrow$ FusionHead(fusion_strategy)
16:                $F_{fused,train} \leftarrow$ combine$(F_{time,train}, F_{spec,train})$
17:                $h_{fusion}$.train$(F_{fused,train}, y_{train})$
18:                $Acc_{fusion} \leftarrow h_{fusion}$.evaluate(combine$(F_{time,test}, F_{spec,test}), y_{test})$
19:                Append $\{..., \text{fusion\_strategy}, Acc_{fusion}\}$ to $R_2$
20:            **end for**
21:        **end for**
22:    **end for**
23: **end for**
24: **Output:** Results $R_2$

---

**Task 2 Workflow**   The workflow for the deep learning fusion experiments (Task 2) is detailed in Algorithm 2. For each model architecture, we first train the base model end-to-end, then extract its penultimate layer features, and subsequently train and evaluate two fusion heads (Concatenation and Gating) using these features combined with spectral information.

---

**Algorithm 3** Pilot Study: Meta-Classifier Workflow

---

1: **Input:** Task 1 Results $R_1$, Meta-Features $MF$
2: Filter $R_1$ for $C = C_{\text{LGBM}}$ and $\Phi = \Phi_{\text{Py-ROCKET}}$
3: Create meta-dataset $\mathcal{D}_{meta}$ by merging filtered $R_1$ with $MF$
4: **for** each row in $\mathcal{D}_{meta}$ **do**
5:     $y_{meta} \leftarrow 1$ if $Acc_{\text{Pruned-KMeans}} > Acc_{\text{Base}}$ else 0
6: **end for**
7: $X_{meta} \leftarrow \mathcal{D}_{meta}[\text{'SNR\_dB', 'Spectral\_Entropy'}]$
8: $M_{meta} \leftarrow$ DecisionTreeClassifier(max_depth $= 1$)
9: $M_{meta}$.fit$(X_{meta}, y_{meta})$
10: **Output:** Learned rule from trained $M_{meta}$

---

**Pilot Study Workflow**   Algorithm 3 specifies the procedure for our pilot study (Section 6). It describes the creation of the meta-dataset by combining experimental results with pre-calculated meta-features, and the subsequent training of a simple decision tree to learn a strategy selection rule.

## A.3 LLM ASSISTANCE DISCLOSURE

Consistent with the conference's transparency policy, we disclose that a Large Language Model (LLM) was utilized as an assistive tool during the preparation of this work. Its application was focused on two distinct support functions: refining the manuscript's prose and accelerating the code debugging cycle.

**Manuscript Preparation** An LLM was employed to improve the linguistic quality of the Abstract and Introduction. This process consisted of iterative, sentence-level prompts designed to enhance the clarity, conciseness, and overall fluency of the text.

**Code Development** During the experimental phase, the LLM functioned as an interactive debugging aid. When encountering software errors, we provided the model with the problematic code segment accompanied by its full traceback and error message. The model was then queried to diagnose the root cause and suggest potential corrections, which streamlined the development process.

It is emphasized that the LLM's role was exclusively that of a productivity tool for language enhancement and code troubleshooting. The core scientific contributions of this paper—including the initial hypothesis, the design of experiments, and the interpretation of results—are solely the original work of the authors.

