# OpenReview forum: "Feature Synergy and Interference: An Analysis for Time-Series Classification"
_ICLR.cc/2026/Conference — ICLR 2026 Conference Withdrawn Submission_

### Official Review · Reviewer_mWE6 · 2025-10-29

**Soundness:** 3
**Presentation:** 3
**Contribution:** 3
**Rating:** 4
**Confidence:** 2

**Summary:**

The paper argues that TSC research should transition from a model-centric to a feature-centric paradigm. By analyzing feature extractors like ROCKET and deep learning models (CNN, RNN, Transformer, Mamba), the authors introduce the notions of feature synergy and feature interference, showing that performance bottlenecks arise from complex interactions between feature noise and classifier capacity. They also find that time–frequency feature fusion behaves differently across model architectures and dataset characteristics. In addition, they propose an adaptive framework that dynamically selects analysis strategies based on dataset properties (e.g., SNR, spectral entropy).

**Strengths:**

- This work argues for a feature-centric perspective, offering a fresh and insightful direction for TSC.
- The authors quantitatively demonstrate how feature interactions manifest differently across model architectures and datasets, providing solid empirical evidence for these theoretical ideas.
- The pilot study showing how dataset meta-features can guide optimal pipeline selection is an innovative proof-of-concept.

**Weaknesses:**

- The pilot study used a small meta-dataset, but the paper does not analyze its statistical limitations or discuss how future work will overcome them.
- Mamba model training failures are reported without detailed analysis (architecture, data regime, optimization, etc.).
- Insufficient justification of design choices: Using the penultimate layer for time-domain features, selecting exactly 50 frequency bands via ANOVA F-test, Unspecified architecture/activations of the gating MLP, and how it aligns feature dimensions.
- Limited theoretical grounding of ‘feature synergy’ and ‘feature interference’
- Narrow dataset coverage (mostly ETT and a subset of UCR/UEA), restricting generalizability across domains (healthcare, finance, industrial control, etc.).
- Scope confined to classification; no evaluation on other TS tasks (forecasting, anomaly detection, etc.).

**Questions:**

- See weakness above.
- What are the root causes of Mamba’s training failures—are they due to architecture design, data characteristics, or optimization instability?
- Could you provide loss curves or gradient stability plots to support the analysis of Mamba’s instability compared with Transformer models?
- How would feature synergy and interference differ across these other tasks (forecasting, anomaly detection, clustering, etc.)?

---

### Official Review · Reviewer_p3Y7 · 2025-10-30

**Soundness:** 2
**Presentation:** 3
**Contribution:** 1
**Rating:** 4
**Confidence:** 4

**Summary:**

This paper presents a feature-centric analysis framework for time-series classification, shifting attention from model architectures to feature interactions. Through two systematic studies, feature pruning in high-dimensional extractors and time–frequency fusion across nine deep models, the authors reveal how “feature synergy” can enhance performance while “feature interference” can degrade it. By correlating these effects with dataset meta-features, the study provides empirical evidence for data-driven adaptation strategies in TSC. Overall, the work offers a new analytical perspective with experimental validation, though its scope and methodological depth remain limited.

**Strengths:**

- The paper departs from the conventional model-centric paradigm by focusing on feature synergy and feature interference across a diverse range of time-series classification models. This conceptual shift provides a fresh analytical framework that could inspire future research on model interpretability and adaptive architecture design.
- The authors conduct two carefully structured studies (feature pruning analysis and deep model fusion analysis) across both shallow and deep architectures. The experiments are reproducible, include proper baselines, and are evaluated on multiple datasets with statistical robustness.
- By introducing dataset-level meta-features such as Signal-to-Noise Ratio and Spectral Entropy, the authors bridge empirical performance with interpretable data characteristics.

**Weaknesses:**

- The related work section is insufficiently comprehensive. The authors entirely overlook recent progress on Large Language Models (LLMs) for time-series classification, which have shown promising capabilities in feature extraction and generalization. The omission of this line of research weakens the paper’s motivation and makes its claimed contributions less convincing, as it appears to be deliberately avoiding comparisons with the rapidly expanding literature on LLMs.
- The study considers only two types of features—time-domain and frequency-domain—while ignoring richer modalities such as semantic features derived from natural language. If the authors aim to establish a general feature-centric framework, they should discuss or experiment with multi-modal extensions, including LLM-driven feature representations. Overall, the work lacks depth and does not reflect sufficient theoretical or empirical exploration of broader feature spaces.

- The methods used mainly rely on older models, such as ROCKET (2019) in Task 1, without explaining why more advanced feature-extraction or fusion methods from recent years were not considered. Likewise, the models employed in Task 2 (like LSTM and ResNet) are general neural architectures not originally designed for time-series classification. This choice influences the evaluation, as even small feature adjustments could surpass these basic baselines. The authors should check if their pruning or fusion strategies also enhance performance on state-of-the-art TSC models, which would strengthen the results.

- The claimed contribution—analyzing pruning and fusion strategies—remains confined to a single downstream task (classification). The findings would be far more valuable if positioned within the broader context of representation learning. Moreover, the paper does not compare its performance with recent state-of-the-art TSC methods, leaving unclear whether the proposed strategies actually yield competitive results within the field.

- Some critical references are absent. For example, the ExchangeRate dataset mentioned around line 198 lacks a citation or source explanation. Such omissions reduce reproducibility and clarity.

- The core concepts of feature synergy and feature interference are not formally defined, relying instead on empirical performance trends. In Section 3.1, the authors apply three pruning strategies—Base, Pruned, and PCA—without explaining the rationale for choosing these specific methods or proposing new, task-specific variants. Similarly, in the fusion analysis, only concatenation and gating are explored, while other plausible operations (e.g., additive fusion, downsampling, resampling) are ignored. This weakens the methodological novelty and interpretability of the study.

- The experiments on feature pruning and fusion are not diverse enough to convincingly support the paper’s conclusions. More datasets, architectures, or ablation analyses are needed to validate the generality of the proposed framework.

**Questions:**

- In Research Question 2, the study focuses exclusively on fusing time-domain and frequency-domain features. Could the authors clarify why other potentially informative modalities (e.g., multi-modal physiological, spatial, or contextual features) were not considered? Would the proposed framework be applicable to broader fusion scenarios?

- Could the observed feature interference phenomenon be attributed to overlapping feature subspaces—for example, when frequency-domain features are linearly dependent on time-domain representations? If so, have the authors explored any quantitative metrics to measure this dependence or redundancy?

- The Mamba model consistently fails on most datasets. Does this indicate an inherent limitation of the architecture for short-sequence classification tasks? If its performance were restored through hyperparameter tuning, would feature interference still be observed, or is the failure purely due to optimization issues?

- The study focuses exclusively on time-series classification. How do the authors expect the observed phenomena of feature synergy and feature interference to generalize to other important tasks, such as forecasting, segmentation, or anomaly detection?

- The notions of synergy and interference are primarily defined empirically through gains or losses in accuracy. Could the authors formalize these concepts more rigorously?

---

### Official Review · Reviewer_dToi · 2025-10-31

**Soundness:** 3
**Presentation:** 3
**Contribution:** 3
**Rating:** 6
**Confidence:** 3

**Summary:**

This study challenges the "one-size-fits-all" model-centric paradigm in TSC, advocating a feature-centric perspective. Via controlled experiments, it analyzes feature spaces of representative TSC models (ROCKET, Transformers, Mamba). For high-dimensional extractors, performance bottlenecks (redundancy/noise) depend on datasets and classifier capacity—pruning denoises for complex non-linear classifiers, while linear models may need full features. Evaluating time-frequency fusion across 9 deep architectures, it reveals feature synergy (fusion boosts performance) and interference (fusion degrades it), with optimal strategies tied to dataset properties and model biases. It also links strategy effectiveness to dataset meta-features (SNR, spectral entropy) and validates an adaptive framework via a pilot study.

**Strengths:**

1. The feature-centric perspective breaks from TSC’s long-standing model-centric focus, filling a critical gap and pointing to a new research direction for robust, dataset-aware solutions.
1. Rigorous experiments (diverse models/datasets, repeated runs, clear baselines) ensure reliable results, while meta-feature correlation adds scientific depth to findings.
1. Specific conclusions (e.g., pruning for non-linear classifiers on noisy data) provide actionable guidance for TSC model design and optimization.
1. The pilot study validates adaptive framework feasibility, bridging theoretical analysis to practical application for automated TSC pipelines.

**Weaknesses:**

1. The adaptive framework relies on a small meta-dataset (LightGBM + Py-ROCKET across 4 datasets), and results are limited to classification, restricting generalizability to other TSC tasks.
1. Key methods like "Pruned-KMeans (unsup.)" lack step-by-step technical explanations; Mamba’s poor performance is not checked via hyperparameter tuning, leaving ambiguity.

**Questions:**

1. For the "Pruned-KMeans (unsup.)" strategy, could you detail how features are clustered and selected for pruning, such as distance metrics or cluster number determination?
1. For the spectral feature selection in time-frequency fusion, could you explain how the 50 most informative bands are prioritized beyond ANOVA F-test, such as handling overlapping band information?

**Details Of Ethics Concerns:**

1. The study mainly uses 4 datasets (ETTh1, ETTm1, ExchangeRate, Weather) — given the diversity of real-world TSC scenarios (e.g., industrial periodic data, medical irregular signals), will limited dataset types restrict conclusion generalizability, and have you considered testing on more diverse datasets (e.g., UCR/UEA with varied temporal patterns)?
2. In Task 2, Mamba’s poor performance is attributed to architectural bias, but there’s no comparison with its performance on standard TSC benchmarks or ablation on input features. Could this lead to misjudging if low performance comes from architecture or setup (e.g., hyperparameters)? For "feature interference" (e.g., GRU on ExchangeRate), no control experiment tests if optimizing spectral features (e.g., wavelet denoising) alleviates interference — why?
3. The adaptive framework’s pilot only uses LightGBM + Py-ROCKET. For other pairs (e.g., RidgeClassifierCV + sk-MINIROCKET with different strategy preferences), has the SNR-based rule been tested? If not, how to adjust it for broader applicability?

I'll raise my score if my concerns are addressed.

---

### Official Review · Reviewer_83Se · 2025-11-02

**Soundness:** 2
**Presentation:** 3
**Contribution:** 2
**Rating:** 4
**Confidence:** 2

**Summary:**

This paper argue that the time series classification (TSC)'s performance, highly depends on data intrinsic feature synergy and inference, rather than using more and more advanced and complex neural network architectures. To support this, the author conduct two experiments. The 1st experiment is to study ROCKET-like feature space. The performance bottleneck in high-dimensional feature spaces (e.g., ROCKET) is not solely redundancy but a complex interaction between feature noise and the capacity of the downstream classifier. For high-capacity, non-linear classifiers (like LightGBM), feature pruning acts as a critical de-noising step on noisy datasets, leading to significant performance gains. Conversely, simpler linear models often benefit more from aggressive dimensionality reduction like PCA or the full feature set. The 2nd task is to evaluate fusion strategies across nine diverse deep learning models. This shows feature synergy and feature interference. The optimal fusion strategy is non-universal, being intricately linked to both the model's architectural biases and the intrinsic properties of the dataset. The paper also proposes an adaptive framework that uses some meta-features to select best strategy for the TSC task.

**Strengths:**

1. The carefully designed task is intereating, with intuitive examples to show the relationship between dataset-features and model-capacities.
2. The paper provides an example to distinguish different kinds of feature fusion, and how to establish a baseline (barycenter approach). This can give researches a standard practice to deal with TSC problems.

**Weaknesses:**

After reading the paper, I feel the paper's contribution to the community is subtle, and this is usually the main issue with this type of analysis papers. It is widely known, and in practice, that feature extraction and interaction is a very critical step to solve complex problems like time series. This paper only verifies this known point, but lack of providing a way to solve the problem. The adaptive framework proposed in this paper is very high-level and primal, which is hard to act as some guideline to solve TSC problem. In addition, the number of datasets is very limited in the paper, which is hard to validate the claim that the authors proposed.

**Questions:**

Does any of the claim consider the impact of data richness? Do they stand the same when dataset is small vs large?

---

### Note · Authors · 2025-11-12

**Comment:**

Dear Program Chair, Area Chairs, and Reviewers,

We are deeply grateful for your thorough review and invaluable feedback on our submission. We sincerely appreciate the time, expertise, and constructive criticism you dedicated to evaluating our work. Your insightful comments have significantly clarified areas requiring refinement, and we fully concur with your suggestions for strengthening the technical rigor and clarity of the paper.

In light of your recommendations, we are committed to revising the manuscript comprehensively to address all points raised. The revised version, incorporating every suggested improvement, will be submitted within the standard revision period.

Thank you for your exceptional guidance, which has been instrumental in elevating the quality and impact of our research. We are truly honored to have your expertise inform this important work.

With utmost respect

**Withdrawal Confirmation:**

I have read and agree with the venue's withdrawal policy on behalf of myself and my co-authors.